# Nurses' work-related stress and associated factors in governmental hospitals in Harar, Eastern Ethiopia: A cross-sectional study

Yohannes Baye[1]*, Tesfaye Demeke[2], Nigusie Birhan[2], Agumasie Semahegn[1], Simon Birhanu[1]

1 School of Nursing and Midwifery, College of Health and Medical Sciences, Haramaya University, Harar, Ethiopia, 2 School of Nursing and Midwifery, College of Medicine and Health Sciences, University of Gondar, Gondar, Ethiopia

* yohannesbaye21@gmail.com

## Abstract

### Introduction

Work-related stress causes poor quality of nursing care and increases the risk of medical errors. Research evidence is so limited to nurses' work-related stress in eastern Ethiopia. Therefore, this study aimed to assess work-related stress and associated factors among nurses working in governmental hospitals in Harar, Eastern Ethiopia.

### Methods

Institution-based quantitative cross-sectional study was conducted among 367 nurses from 15[th] to 30[th] March, 2015. Simple random sampling technique was applied to recruit study participants. Data were collected using structured self-administered questionnaire. Descriptive statistics, bivariate and multivariate logistic regressions were carried out. The statistical association was declared using adjusted odds ratio at 95% confidence interval (CI) and P-value of less than 0.05.

### Results

A total of 398 study participants were involved in the study, and the response rate was 92.2% (367/398). More than half of 202(55%) of the participants were males. One third (33.8%, n = 124) of study participants' age ranged between 26 to 34 years. The prevalence of work-related stress in the current study was 66.2%. Nurses, who reared child (AOR = 2.1, 95% CI: 1.2, 3.7), working in intensive care units (AOR = 4.5, 95% CI: 1.4, 17.7), work on rotation (AOR = 2.5, 95% CI: 1.4, 4.4), and nurses who had a chronic medical illness (AOR = 2.6, 95% CI: 1.2, 5.7) were significantly associated with nurses' work-related stress.

### Conclusion

Two-thirds of nurses who were working at government hospitals had work-related stress. Work-related stress was associated with child-rearing, working units, work on rotation, and chronic medical illness. We suggested the hospital's administration, and other concerned

**Data Availability Statement:** All relevant data are within the manuscript.

**Funding:** The author(s) received no specific funding for this work.

**Competing interests:** The authors have declared that no competing interests exist.

**Abbreviations:** AH, Army Hospital; AOR, Adjusted odds ratio; COR, Crude odds ratio; CI, Confidence interval; ENSS, Expanded nurses stress scale; HUSTH, Haramaya University specialized Teaching Hospital; ICU, Intensive care unit; IQR, Interquartile range; JU, Jugel Hospital; NSS, Nurses stress scale; OPD, Outpatient department; PH, Police Hospital.

stakeholders should design a strategy to undertake necessary measures such as hiring more nurses to minimize workload and rescheduling work shift to alleviate work-related stress among nurses.

## Introduction

World Health Organization has considered stress as a global epidemic, which has recently observed to be associated with 90% of visits to physicians [1]. Work-related stress is one of the most important workplace health risks for employees worldwide [2]. Work-related stress results in substantial costs to employees and organizations [3–5], related to employees absenteeism and turnover, decreased productivity, physical illness, poor quality of health care services, and increase risk of medical errors [6].

Globally, the costs of work-related stress are estimated to be $5.4 billion each year, which is one of the most frequently reported occupational health problems [7]. Stress is derived from the word "Stringi", which means "to be drawn tight". Stress can be defined as a physical or psychological stimulus that can produce mental tension or physiological reactions that may lead to illness [8].

Work-related stress has recognized as the main challenge for the nursing profession throughout the world and has negative emotional, physical, and psychological effects on the nurse [9, 10]. Research evidence demonstrated that nurses suffer from high levels of work-related stress are threatening their health, patients' lives, compromise the quality of nursing care, and increasing the cost of health care [11]. Excessive occupational stress has been found to reduce the quality of nursing care [12]. If a nurse is stressed, it is difficult to give holistic nursing care to patients which may increase patient mortality rate [13, 14].

Nursing job (work-related) stress can be defined as the physical and emotional reactions that occur when the nurses' abilities and resources imbalance with the demands and requests of their work [15, 16]. Occupational stress, job stress, organizational stress, and work-related stress are interchangeably used terms [17].

The nature of the nursing profession and the health care system are some of the contributors to work-related stress [18, 19]. Research findings have indicated that the sources of occupational stress, its levels, and effects vary depending on local factors such as the nature of work, work setting, and cultural orientation. Thus, occupational stress among nurses may have significant differences in different countries due to different work settings and levels of social support [20]. Therefore, identification of the sources and contributing factors to work-related stress is necessary for improving the stress management program in the organization.

In Ethiopia, nurses have been playing a crucial role in the health care delivery system. However, there is limited research evidence regarding work-related stress among nurses, specifically in the study area. Therefore, this study aimed to assess work-related stress and associated factors among nurses working in governmental hospitals in Harar, Eastern Ethiopia.

## Methods and materials

### Study setting and design

An institution-based quantitative cross-sectional study was carried out among nurses working in governmental hospitals in Harar City, Harari Regional State, Eastern Ethiopia from March 15 to 30, 2015. Harar city is the political capital of the Harari Regional State. There are a total

of four governmental hospitals (2 military and 2 public hospitals) in Harar. Namely, the 2 military hospitals are Police Hospital (PH) and Army Hospital (AH). Likewise, the 2 public hospitals are Haramaya University Specialized Teaching Hospital (HUSTH) and Jugel Hospital (JH). The military hospitals are for the police, army, and their families. On the other hand, public hospitals have been serving the general public. This study was conducted in the four governmental hospitals in Harar. There were a total of 446 nurses working in these four governmental hospitals during the study period. Out of these, HUSTH comprised 244 nurses and JH had 97 nurses. Similarly, there were 55 nurses in AH and 50 nurses in PH working in different departments.

## Participants and sampling procedure

All nurses (446) who were working in government hospitals in the Harari region were the source population. The sample size was calculated using a single population proportion formula at 95% significance level, by considering the proportion of work-related stress (37.8%) in Addis Ababa [21] with a 5% margin of error, and adding 10% non-response rate. Then, the total sample size was 398 nurses. The calculated sample size was proportionally allocated to the size of the four government hospitals, HUSTH, JH, AH, and PH constituted 217, 87, 49, and 45 participants respectively. The sampling frame was constructed for each hospital separately by taking the nurses list from the human resource management department of each hospital. Then, study participants were selected from respective hospital units using a simple random sampling technique.

## Study population

All nurses who were fulltime employees of Harari Regional State governmental hospitals were included in the study. Nevertheless, nurses who were seriously ill or unable to give responses due to this illness during the data collection period were excluded.

## Data collection tools and procedures

Data were collected using a structured and pre-tested self-administered questionnaire. The tool consisted of the socio-demographics and questions related to the work environment, substance use, illness, and stress. Work-related stress was assessed based on the Expanded Nursing Stress Scale (ENSS). ENSS is an expanded and updated Nursing Stress Scale developed by French SE and her colleagues The Scale comprised of 57 items in nine subscales. These are; death and dying (7 items); conflict with physicians (5 items); inadequate emotional preparation (3 items); problems relating to peers (6 items); problems relating to supervisors (7 items); workload (9 items); patients and their families (8 items); discrimination (3 items), and treatment uncertainty (9 items) [22].

Participants were asked to indicate the frequency of work-related stress using a 4-point response scale. Response options are 'never stressful' (1), 'occasionally stressful' (2), 'frequently stressful' (3), and 'always stressful' (4). The category "not applicable" was scored as zero (0). To compute the total stress score, we added together with the scores on all 57 items. To measure scores on specific subscales, the appropriate items should be added together and the higher the score, the greater the frequency of stress on any subscale. The higher the score, the more the respondent agrees that the situation was stressful. Work-related stress is determined by computing the mean of all 57 items for every study participant. In this study, the mean values 2 and above were assumed to indicate Work-related stress. The overall ENSS reliability with Cronbach's alpha ($\alpha$) was 0.96. The individual subscale reliability ranged from $\alpha = 0.88$ (problems with supervisors) to $\alpha = 0.65$ (discrimination) [22]. In the current study, the internal

**Table 1. Reliability test of the expanded nursing stress scale, 2015.**

| Stressors | Number of items | Scale Mean | Scale Variance | Cronbach's Alpha |
|---|---|---|---|---|
| Work load | 9 | 106.64 | 703.14 | 0.85 |
| Death and dying | 7 | 107.84 | 749.96 | 0.84 |
| Uncertainty concerning treatment | 9 | 108.53 | 737.29 | 0.84 |
| Patient and family | 8 | 109.34 | 716.60 | 0.83 |
| Problems with supervisors | 7 | 111.58 | 735.06 | 0.83 |
| Problems with peers | 6 | 115.17 | 764.26 | 0.83 |
| Conflict with physicians | 5 | 116.42 | 824.13 | 0.84 |
| Inadequate emotional preparation | 3 | 120.79 | 872.15 | 0.85 |
| Discrimination | 3 | 122.73 | 864.23 | 0.86 |
| Total | 57 | | | **0.86** |

consistency (reliability) of the items with Cronbach's alpha was found to be 0.86 for the overall scales (Table 1).

## Operational definitions

**Work-related stress** is defined as the physical and emotional reactions that occur when the nurses' abilities and resources imbalance with the demands and requests of their work [15, 16].

**Nurses' work-related stress** was rated from 1 (never stressful) to 4 (always stressful); the mean score of 2 or higher of 57 items was considered as work-related stress [22].

**Substance use.**   Use or consumption of any substance such as alcohol, chat, cigarettes, shisha, and hashish, regardless of the amount and frequency of use for the past 3 months [21].

**Job satisfaction.**   Job Satisfaction was defined as whether respondents like (satisfied) or dislike (dissatisfied) their jobs [21]. In this study, nurses who answered "yes" were assumed to be satisfied with their job.

**Chronic medical illness.**   Chronic medical disease was categorized as 'yes' if hypertension, cardiovascular disease, AIDS, diabetes mellitus, or arthritis had been diagnosed [23].

**Child-rearing.**   Child rearing was categorized as 'yes' if the person is parenting or taking care of a child/children.

## Quality control measures

Training of the data collectors and supervisors was performed by the investigators. Pretesting was undertaken on 20 respondents before the actual data collection at other hospital and some questions were modified based on the participant's response. Each filled questionnaires were checked thoroughly for its completeness and consistency, and necessary feedback was offered to a data collector in the next morning.

## Data processing and statistical analysis

Collected data were entered into EpiInfo7 and then exported into SPSS (20) software for analysis. Data exploration was performed by executing frequency distribution with a normal curve and box plot to identify outliers and some data anomalies. Data cleaning was carried out as necessary. Descriptive statistics were carried out to compute proportion, frequency, mean and standard deviations. Binary logistic regression was computed to identify association between independent variables with nurses' work-related stress. Variables that had a *p*-value of less than 0.20 in bivariate analysis were in cluded in the multivariate analysis to control for possible confounders. Finally, 95% confidence interval, adjusted odds ratio, and P-values less than 0.05

was used to determine the statistically significant association between independent variables and nurses' work-related stress.

## Ethical approval and consent to participant

Ethical clearance was obtained from the ethical review committee of the Department of Nursing, University of Gondar. A formal letter of cooperation was written to Harari People's National Regional State Health Bureau and respective hospitals. Informed written consent was obtained from each study participant. The purpose of the study was explained to each study participant. All information obtained from the study subject would be kept confidential anonymously.

## Results

### Socioeconomic and demographic characteristics of study participants

A total of 398 nurses have participated in the study, and the response rate was 92.2% (n = 367). More than half of 202(55%) of the participants were males. The median age was 28 (interquartile range (IQR) = 24–38) years. One third (33.8%, n = 124) of study participants' age ranged between 26 to 34 years. More than half (58.6%, n = 215) of participants were belonging Ethiopian Orthodox Christianity. Most of the participants were the Amhara ethnicity (46.6%, n = 171). Nearly half (48.2%, n = 177) of participants were married. More than half (51.8%, n = 190) of nurses have not reared the child. More than half (58.3%, n = 214) of study participants had a Bachelor of Science Degree in Nursing (Table 2).

**Table 2. Socio-demographic characteristics of nurses working in governmental hospitals in Harar, Eastern Ethiopia, 2015 (n = 367).**

| Variables | Categories | Frequency | % |
|---|---|---|---|
| Age (Years) | ≤ 25 | 123 | 33.5 |
| | 26–34 | 124 | 33.8 |
| | 35–44 | 93 | 25.3 |
| | ≥45 | 27 | 7.3 |
| Sex | Male | 202 | 55 |
| | Female | 165 | 45 |
| Religion | Orthodox | 215 | 58.6 |
| | Muslim | 81 | 22.1 |
| | Protestant | 65 | 17.7 |
| | Catholic | 6 | 1.6 |
| Ethnicity | Amhara | 171 | 46.6 |
| | Oromo | 107 | 29.2 |
| | Tigre | 33 | 9 |
| | Harari | 26 | 7.1 |
| | Guraghe | 20 | 5.4 |
| | "Others" | 10 | 2.7 |
| Marital status | Married | 177 | 48.2 |
| | Single | 168 | 45.8 |
| | Divorced | 10 | 2.7 |
| | Separated | 8 | 2.2 |
| | Widowed | 4 | 1.1 |
| Child rearing | No | 190 | 51.8 |
| | Yes | 177 | 48.2 |

(*Continued*)

**Table 2.** (Continued)

| Variables | Categories | Frequency | % |
|---|---|---|---|
| **Level of Education** | BSc | 214 | 58.3 |
| | Diploma | 153 | 41.7 |
| **Monthly salary (in Birr)** | 1200–1560 | 32 | 8.7 |
| | 1561–2100 | 67 | 18.3 |
| | 2101–2620 | 83 | 22.6 |
| | >2620 | 185 | 50.4 |

"Others" include; Kembata, Somali and Wolayta.

## Work environment and behavioral characteristics of study participants

The study participants were working in Haramaya University specialized Teaching Hospital 200 (54.4%), Jugol hospital 80(21.7%), Army hospital 45(12.3%), and Police hospital 42 (11.4%). Nurses who were working in outpatient departments, surgical wards, medical wards, and intensive care units were 15.0%, 14.4%, 13.6%, and 12.8% respectively. Two-thirds (67.0%, n = 246) of the nurses worked rotating shift and reported having ≤7 years of work experience with the median year of nurses' work experience was 4 (IQR = 1 to 11) years. Approximately three-fourth (71.7%, n = 263) of nurses worked 35 to 50 hours per week. Almost half (51.5%, n = 189) of nurses reported being dissatisfied with their jobs (Table 3).

## Prevalence of nurses' work-related stress

The total ENSS score for nurses' ranged from 52 to 194 and the mean ranged from 0.9 to 3.4. The prevalence of nurses' work-related stress was 66.2%. Nurses' workload was the most

**Table 3. Work environment and behavioral characteristics of Nurses in Harar, Eastern Ethiopia, 2015 (n = 367).**

| Variables | Categories | Frequency | % |
|---|---|---|---|
| **Work unit** | Medical ward | 50 | 13.6 |
| | Surgical ward | 53 | 14.4 |
| | Pediatrics ward | 29 | 7.9 |
| | Maternity ward | 41 | 11.2 |
| | Psychiatry ward | 13 | 3.5 |
| | Emergency unit | 40 | 10.9 |
| | Intensive care unit | 47 | 12.8 |
| | Operation room | 39 | 10.6 |
| | Outpatient department | 55 | 15.0 |
| **Work shift** | Rotating | 246 | 67.0 |
| | Fixed | 121 | 33.0 |
| **Job rank** | Basic nurse | 331 | 90.2 |
| | Head nurse | 36 | 9.8 |
| **Working hours per week** | 35–50 | 263 | 71.7 |
| | 51–65 | 76 | 20.7 |
| | 66–80 | 24 | 6.5 |
| | ≥81 | 4 | 1.1 |
| **Job satisfaction** | No | 189 | 51.5 |
| | Yes | 178 | 48.5 |

(*Continued*)

**Table 3.** (Continued)

| Variables | Categories | Frequency | % |
|---|---|---|---|
| **Work experience** | ≤ 7 years | 250 | 68.1 |
| | 8–14 years | 46 | 12.5 |
| | 15–21 years | 44 | 12.0 |
| | 22–28 years | 21 | 5.7 |
| | 29–38 years | 6 | 1.6 |
| **Having a chronic medical illness** | No | 302 | 82.3 |
| | Yes | 65 | 17.7 |
| **Substance use** | No | 304 | 82.8 |
| | Yes | 63 | 17.2 |

frequently reported stressful sub-scale (mean = 20.7) followed by death and dying (mean = 19.5) (Table 4).

## Factors associated with nurses' work-related stress

In bivariate analysis nurses' child-rearing, working unit, work shift, chronic medical illness, and work experience were significantly associated with nurses' work-related stress. However, in the multivariate analysis; child-rearing, working unit, work shift, and chronic medical illness showed a significant association with nurses' work-related stress. Nurses who reared children were 2 times more likely to experience work-related stress than those who did not rear child (AOR = 2.1, 95% CI: 1.2, 3.7). Nurses who were working in the intensive care unit were 4.5 times more likely to experience work-related stress than nurses were working in the outpatient department (AOR = 4.5, 95% CI: 1.2, 17.7). The odds of nurses who were working in psychiatry wards were 90% less likely report work-related stress than nurses who were working in the outpatient department (AOR = 0.1, 95% CI: 0.0, 0.6). Nurses who were working on rotating shifts were 2.5 times more likely to experience occupational stress than those who were working on fixed shifts (AOR = 2.5, 95% CI: 1.4, 4.4). Nurses who had chronic medical illnesses were 2.6 times more likely to suffer from work-related stress than nurses who had no known chronic illness (AOR = 2.6, 95% CI: 1.2, 5.7) (Table 5).

**Table 4. Means and standard deviations of the stressors among nurses in Harar, Eastern Ethiopia, 2015 (n = 367).**

| Stressors | Number of items | Mean | SD |
|---|---|---|---|
| Work load | 9 | **20.7** | 6.9 |
| Death and dying | 7 | **19.5** | 5.4 |
| Uncertainty concerning treatment | 9 | 18.8 | 6.1 |
| Patient and family | 8 | 18.0 | 5.7 |
| Problems with supervisors | 7 | 15.8 | 5.3 |
| Problems with peers | 6 | 12.2 | 4.4 |
| Conflict with physicians | 5 | 11.0 | 3.8 |
| Inadequate emotional preparation | 3 | 6.6 | 2.7 |
| Discrimination | 3 | 4.7 | 3.7 |
| Total | 57 | | |

"SD" = Standard deviation.

**Table 5. Work-related stress and associated factors among nurses in Harar, Eastern Ethiopia, 2015.**

| Variables | Categories | Work-related stress | | COR (95% CI) | AOR(95% CI) |
|---|---|---|---|---|---|
| | | Non- stressed | Stressed | | |
| Child rearing | No | 81 | 109 | 1.00 | 1.00 |
| | Yes | 43 | 134 | **2.3(1.5,3.6)**[a] | **2.1(1.2,3.7)**[b] |
| Work unit | OPD | 22 | 33 | 1.00 | 1.00 |
| | Surgical ward | 21 | 32 | 1.0(0.5,2.2) | 0.5(0.2,1.1) |
| | Medical ward | 19 | 31 | 1.1(0.5,2.4) | 0.6(0.2,1.5) |
| | ICU | 3 | 44 | **9.8(2.7,35.4)**[a] | **4.5(1.2,17.7)**[b] |
| | Maternity ward | 18 | 23 | 0.9(0.4,1.9) | 0.6(0.3,1.5) |
| | Emergency ward | 7 | 33 | **3.1(1.2,8.4)**[a] | 2.0(0.7,5.8) |
| | Operation room | 16 | 23 | 1.0(0.4,2.2) | 0.6(0.2,1.4) |
| | Pediatrics ward | 8 | 21 | 1.8(0.7,4.6) | 0.9(0.3,2.7) |
| | Psychiatry ward | 10 | 3 | **0.20(0.0,0.8)**[a] | **0.1(0.0,0.6)**[b] |
| Work shift | Fixed | 58 | 63 | 1.00 | 1.00 |
| | Rotating | 66 | 180 | **2.5(1.6,4.0)**[a] | **2.5(1.4,4.4)**[b] |
| Chronic medical illness | No | 114 | 188 | 1.00 | 1.00 |
| | Yes | 10 | 55 | **3.3(1.6,6.8)**[a] | **2.6(1.2,5.7)**[b] |
| Work experience (in a year) | 29–38 | 4 | 2 | 1.00 | 1.00 |
| | ≤7 | 95 | 155 | 3.3(0.6,18.2) | 2.4(0.4,15.6) |
| | 8–14 | 6 | 40 | **13.3(2.0,89.3)**[a] | 7.0(0.9,53.6) |
| | 15–21 | 12 | 32 | 5.3(0.9,33.0) | 2.5(0.3,17.6) |
| | 22–28 | 7 | 14 | 4.0(0.6,27.4) | 2.6(0.3,21.1) |

N.B.

[a] Significant at $p < 0.20$

[b] Significant at $p < 0.05$

## Discussion

This study determined the level of nurses' work-related stress. The prevalence of work-related stress was 66.2%. This finding is much higher as compared to a study in Addis Ababa (37.8%) [21]. The difference might be due to the tool used in Addis Ababa which was NSS. This study was also much higher as compared to a study conducted in Malacca (28.3%) [24]. The possible reason for the difference might be due to the study setting in which the Malacca study was conducted only in maternal and child health clinics. Similarly, the number of nurses reporting work-related stress in this study was higher than the stress in Kuala Lumpur (24.6%) [25], Brazil (23.6%) [26] and Taiwan (17.2%) [27]. The difference between these studies and the current study might be due to the tools used and study settings. However, the prevalence of the current study was lower when compared to studies done in Malaysia (100%) [28], Egypt (92%) [29], China (86.9%) [23] and India (73.5%) [30]. The possible reason for this variation might be.due to the tools used and study setting. Similarly, the current study was lower when compared to a study conducted in Iran which reported that 73.4% of nurses experienced work-related stress [31]. This variation might be due to the study setting since the Iranian study was carried out only in a teaching hospital, whereas the current study included all types of hospitals. The descriptive analysis indicated that the workload was the most stressful subscale for nurses. This might be due to a shortage of staff, extra non-nursing tasks, and less time to accomplish the work and support each other emotionally. Ethiopia, like other developing countries, suffers from a shortage of nurses (2 nurses for 10,000 people) [8] which increases nurses' workload.

This finding was in line with researches reported in Addis Ababa which stated that a greater source of stress for nurses was "not enough staff to adequately cover unit" followed by "not enough time to finish all their nursing tasks [21], South Africa which indicated nurses' stress was caused by "not enough staff to adequately cover unit" [32], and Thailand which was found that heavy workloads caused high work-related stress in nurses [33].

In this study, a significant association was found between working units and work-related stress. Nurses who worked in intensive care units were experienced more stress than other units/wards. This was consistent with studies conducted in Malaysia [28] and Ireland [34] which have revealed that nurses working in ICU were found to have a high level of work-related stress. This might be due to nurses working in intensive care units are busy, caring for critical patients, and have little time to support each other emotionally. This finding was different from a study in Addis Ababa which reported that nurses who worked in medical wards and emergency units were stressed more [21]. The reason might be due to the number of nurses assigned to intensive care units and patient flow in this study area would not be proportioned. Similarly, this finding was inconsistent with a study in Malaysia which showed that respondents working in the department of medicine experienced a higher level of stress compared to those working in the other departments [25].

In this study, a significant association was also found between work shift and work-related stress. Rotating shift nurses were more stressed than fixed shift nurses. This finding was consistent with researches reported in Addis Ababa [21], Egypt [35], and Jordan [36] which have indicated that nurses who worked rotating shifts were more stressed than nurses who worked fixed shifts. Researchers also found a significant association between child-rearing and work-related stress in this study; nurses who reared children were more likely to experience work-related stress than those who did not rear child. This finding was supported by a South Korean study [37] that stated nurses' working stress had significant correlation with parenting.

Even though this research could not differentiate temporal relationships, respondents who reported chronic medical illness were more likely to report work-related stress than those without illness. This finding was consistent with studies conducted in Ethiopia [21], China [23], and research reported by Seyle's [38].

## Strength and limitation of the study

This study had some limitations, where the cross-sectional study design used in this study cannot conclude a temporal relationship between exposure and disease. Since stress is mainly subjective and psychological, the qualitative approach would provide rich and meaningful information about the nurses' experiences with stress and related concepts.

## Conclusion

This study determined the level of work-related stress among nurses working in government hospitals in Harar, Ethiopia. Two-third of nurses working in governmental hospitals had work-related stress. Child rearing, working unit, work shift, and chronic medical illness were statistically significantly associated with nurses' work-related stress. Nurses have a tremendous role in the health care delivery system in Ethiopia. Because nurses worked in a poor resource setting and overflow of patients with communicable and non-communicable health problems, nurses have been facing many challenges in their work environment. In the meantime, nurses experienced work-related stress which may affect the quality of health care service, increase medical errors and resource wastage. We suggested the need for organizational interventions to reduce the effect of this work-related stress. Hiring more nurses might be a potential remedy to minimize workload with increasing clerical staff to reduce non-nursing tasks. Strategies that

reduce stress due to shift work should be considered such as rescheduling. Further study using a mixed-method and analytical design in governmental and private health facilities is recommended to establish a real cause-effect relationship.

## Acknowledgments

We would like to acknowledge study sites' administrative bodies for their kindly cooperation during the data collection period. We also would like to thank data collectors and study participants without whom this could not be realized.

## Author Contributions

**Conceptualization:** Yohannes Baye.

**Data curation:** Yohannes Baye.

**Formal analysis:** Yohannes Baye.

**Investigation:** Yohannes Baye.

**Methodology:** Yohannes Baye, Tesfaye Demeke, Nigusie Birhan, Agumasie Semahegn, Simon Birhanu.

**Software:** Simon Birhanu.

**Supervision:** Yohannes Baye.

**Validation:** Yohannes Baye.

**Writing – original draft:** Yohannes Baye, Simon Birhanu.

**Writing – review & editing:** Tesfaye Demeke, Nigusie Birhan, Agumasie Semahegn.

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
