## [Decision Letter · Decision Letter 0]

10 Oct 2019

PONE-D-19-14479

Nurses’ work-related stress and associated factors in governmental hospitals in Harar, Eastern Ethiopia: A cross-sectional study

PLOS ONE

Dear MR. Baye,

Thank you for submitting your manuscript to PLOS ONE. After careful consideration, we feel that it has merit but does not fully meet PLOS ONE’s publication criteria as it currently stands. Therefore, we invite you to submit a revised version of the manuscript that addresses the points raised during the review process.

We would appreciate receiving your revised manuscript by Nov 24 2019 11:59PM. To enhance the reproducibility of your results, we recommend that if applicable you deposit your laboratory protocols in protocols.io, where a protocol can be assigned its own identifier (DOI) such that it can be cited independently in the future. For instructions see: http://journals.plos.org/plosone/s/submission-guidelines#loc-laboratory-protocols

We look forward to receiving your revised manuscript.

Kind regards,

Sphiwe Madiba, DrPH

Academic Editor

PLOS ONE

Journal Requirements:

https://www.researchgate.net/publication/264888293_Effect_Procrastination_on_Work-Related_Stress

http://www.imedpub.com/articles/occupational-stress-management-among-nurses-in-selected-hospital-in-benin-city-edo-state-nigeria.php?aid=12016

https://www.scirp.org/journal/PaperInformation.aspx?PaperID=91107

The text that needs to be addressed is in the Background section

In your revision ensure you cite all your sources (including your own works), and quote or rephrase any duplicated text outside the methods section. Further consideration is dependent on these concerns being addressed.

4. Please include additional information regarding the survey or questionnaire used in the study and ensure that you have provided sufficient details that others could replicate the analyses. For instance, if you developed a questionnaire as part of this study and it is not under a copyright more restrictive than CC-BY, please include a copy, in both the original language and English, as Supporting Information.  If the original language is written in non-Latin characters, for example Amharic, Chinese, or Korean, please use a file format that ensures these characters are visible.

'We would like to thank the University of Gondar for financial support.'

'The author(s) received no specific funding for this work.'

Additional Editor Comments

The introduction is too brief; there is need for a thorough review of literature to indicate the problem. For example, the authors need to indicate the prevalence and effects on work-related stress on the performance of the organization as well as the nurses in other countries and in Ethiopia. Furthermore, the authors did not justify the reason for conducting the study. The paragraph on the implication of the study can be worked into the justification or introduction.

I agree that the sampling is confusing and needs to be revisited and clarified.

In the discussion, the authors compare their findings to studies conducted elsewhere with little efforts to compare to studies in Ethiopia and other parts of Africa. There is adequate research conducted in this topic in many parts of sub Saharan Africa to support the study findings. It is also not clear why the authors compare their findings to those with different study populations.

The authors state that working in ICU was significantly associated with work related stress, they further stated that the association might be due to the fact that nurses working in intensive care units are busy, caring for critical patients, and have little time to support each other emotionally.  It is important that the authors report what other studies attributed the link between working in ICU and work related stress to.

The author need to summarize and synthesize the findings, for example they report that “*Their finding was consistent with research done in Addis Ababa which reported that nurses who worked rotating shifts were more stressed than nurses who worked fixed shifts (17). Similarly, this finding was consistent with a study in Egypt that revealed nurses working at rotating shift were more stressed than those who were working in the morning shift (18) and in Jordan reported that work shift was the best predictor of nurses’ stress (19)” this should be summarized.* 

The work related stressors assessed through Expanded Nursing Stress Scale (ENSS) are not well discussed; there is only mention of workload as the main stressor.

The authors need to review the conclusion once they revise the discussion and introduction.

Reviewers' comments:

Reviewer's Responses to Questions

**Comments to the Author**

1. Is the manuscript technically sound, and do the data support the conclusions?

Reviewer #1: Yes

Reviewer #2: Yes

2. Has the statistical analysis been performed appropriately and rigorously? 

Reviewer #1: Yes

Reviewer #2: Yes

3. Have the authors made all data underlying the findings in their manuscript fully available?

Reviewer #1: Yes

Reviewer #2: Yes

4. Is the manuscript presented in an intelligible fashion and written in standard English?

Reviewer #1: Yes

Reviewer #2: No

5. Review Comments to the Author

Reviewer #1: Abstract

From conclusion part it is better describing the measures for those significantly associated factors separately rather than reporting like " ....undertake necessary measures on associated factors " ( line 20)

Methods and materials

study settings and design sections lack more information's on ;

Total number of governmental hospitals in the study area (in Harar region) and selection of hospitals is not clear ?

total number of health professionals and nurses as well working during the study time ?

what are the inclusion and exclusion criteria's of the study ? not clearly mentioned or I think it is missed from the manuscript ?

What is your assumption with the outcome variables from the study units or across hospitals ,is it similar or different ?

variable measurement or operational definition Missed.

First, a definition of work related stress would be helpful to the reader.

then, the measurements of your outcome variable work related stress ; "stressed or not stressed " ,should be clearly defined .when did you say "there was nurse's work related stressed ? and/or not stressed ?" .

Hint; how could you measure the tool Expanded Nursing Stress 19 Scale (ENSS) which contains 57 items that assess nine areas of stress having multiple response ?

Similar measurement clarifications may also necessary for other independent variables like ;job satisfaction ,work load ?

What is Cronbach’s alpha coefficient implications and its cut off point for conclusion? (table 1)

Discussion

was the tools used for the study were, the primary justifications for variations of different findings with yours,what are other possible expectations or perspectives behind for the difference can you suggest? (table 2 ,5)

Conclusion

making the general conclusion with the descriptive result is not recommended.so try to remove this sentence "The descriptive analysis showed that the work load was the most frequently stressful sub-scale ."

Reviewer #2: The manuscript has no page numbers. Editor relies on whole article pagination as downloaded from website. Assuming that the Abstract is on page 8.

Please include socio-demographics in abstract results

2. Sampling section not clearly written. Please revisit. Author starts with simple random sampling but ends up with stratified random sampling but provides numbers for each strata. The sampling frame is small (446). Why was a census not considered?

4. There are serious language and typographical errors in the whole manuscript. This needs serious attention.

Keep reporting of numbers to 1 decimal place.

Pg 8 ln 9: …statistics were carried out. For what?

Pg 9 ln 4 and 20. Fix English: change has incurred to incurs and employed to carried out .

6. PLOS authors have the option to publish the peer review history of their article (what does this mean?). If published, this will include your full peer review and any attached files.

Reviewer #1: Yes: Getnet Gedif Engida

Reviewer #2: No

---

## [Author Response · Author response to Decision Letter 0]

6 Jan 2020

Response to Reviewers’ 

First of all, we would like to appreciate the PLOSE ONE Journal Academic Editor and Reviewers for providing us timely and constructive comments to revise our manuscript that suits the PLOS ONE Journal standards and requirements. We are also pleased to submit the revision that responds to each point raised by the academic editor/ reviewers and presented as follows.

 Upon our revision, we have seen that some of the manuscript parts do not meet the PLOS ONE's style requirements. After thoroughly reading the PLOS ONE's submission guidelines we made changes that meet the requirement.

2. We suggest you thoroughly copyedit your manuscript for language usage, spelling, and grammar.

 As much as possible we tried to improve the manuscript for language usage, spelling, and grammar by consulting our colleagues and staff language professionals.

https://www.researchgate.net/publication/264888293_Effect_Procrastination_on_Work-Related_Stress

http://www.imedpub.com/articles/occupational-stress-management-among-nurses-in-selected-hospital-in-benin-city-edo-state-nigeria.php?aid=12016

https://www.scirp.org/journal/PaperInformation.aspx?PaperID=91107

 Thanks for your link and we searched and have seen some minor occurrence of overlapping text especially with the last two papers. The overlapping was on the definition of “stress” and we quoted for that and rephrased the rest text.

4. Please include additional information regarding the survey or questionnaire used in the study and ensure that you have provided sufficient details that others could replicate the analyses.

 As mentioned in the methodology part of the manuscript, the tool (questionnaire) is a structured and pre-tested self-administered questionnaire. The tool consisted of the socio-demographics and ques¬tions related to the work environment, substance use, illness, and stress. 

 Work-related stress was assessed based on the Expanded Nursing Stress Scale (ENSS). ENSS is an expanded and updated Nursing Stress Scale developed by French SE, Lenton R, Walters V, and Eyles J. The ENSS was obtained directly from the author Susan French communicating through email (susan.french@mcgill.ca) and getting permission.

 The Expanded Nursing Stress Scale contained 57 items in nine subscales: death and dying (7 items); conflict with physicians (5 items); inadequate emotional preparation (3 items); problems relating to peers (6 items); problems relating to supervisors (7 items); workload (9 items); patients and their families (8 items); discrimination (3 items) and treatment uncertainty (9 items). Participants were asked to indicate the frequency of work-related stress using a 4-point Likert response scale. Response options were ‘never stressful’ (1), ‘occasionally stressful’ (2), ‘frequently stressful’ (3) and ‘always stressful’ (4). The category “not applicable” was scored as zero (0).

 In order to compute the total stress score, we added together with the scores on all 57 items. In order to measure scores on specific subscales, the appropriate items should be added together and the higher the score, the greater the frequency of stress on any subscale. The higher the score, the more the respondent agrees that the situation was stressful. 

 Work-related stress is determined by computing the mean of all 57 items for every study participant.

 In this study, the mean values 2 and above were assumed to indicate Work-related stress.

 Based on a study done on a random sample of 2,280 nurses in Ontario, Canda, the authors demonstrated the ENSS reliability with Cronbach’s alpha (α = 0.96) and the individual subscale reliability ranged from α = 0.88 (problems with supervisors) to α = 0.65 (discrimination) [31]. 

 In the current study, Cronbach’s alpha coefficient was found to be 0.86 for the overall scales [Table1].

 We removed funding information in our revised manuscript and only included in the Funding Statement section of the online submission form.

6. The introduction is too brief; there is a need for a thorough review of literature to indicate the problem.

 In the revised manuscript we have included the prevalence and effects of work-related stress on the performance of the organization as well as the nurses in Ethiopia and other countries. We also incorporated the justification of conducting this study.

 We revised the methodology part and made some clarification.

 We also incorporated and compared similar studies conducted in Ethiopia and other parts of Africa.

7. The total number of governmental hospitals in the study area (in the Harar region) and the selection of hospitals is not clear?

 There are a total of four governmental hospitals (2 military and 2 public hospitals) in the Harari region. Namely, the 2 military hospitals are Police Hospital (PH) and Army Hospital (AH). Likewise, the 2 public hospitals are Haramaya University Specialized Teaching Hospital (HUSTH) and Jugel Hospital (JH). The military hospitals usually serve the police and army population, on the other hand, the public hospitals serve the general population. 

 The study was conducted in the four Governmental hospitals found in the Harari region. There were a total of 446 nurses working in these four Governmental hospitals during the study period. Out of these, HUSTH comprised 244 nurses and JH had 97 nurses. Similarly, there were 55 nurses in AH and 50 nurses in PH working in different departments.

 The sample size was calculated using a single population proportion formula at 95% significance level, by considering the proportion of work-related stress (37.8%) in Addis Ababa with 5% margin of error, and adding 10% non-response rate. Therefore, the total sample size was 398 nurses.

 A stratified sampling technique was applied considering the difference in the level of work-related stress across hospitals. Therefore, the total sample size was proportionally allocated to the size of the four government hospitals so that HUSTH, JH, AH and PH constituted 217, 87, 49 and 45 participants respectively. The sampling frame was constructed for each hospital separately by taking the nurses list from the human resource management department of each hospital. Then, study participants were selected from respective hospital units using a simple random sampling technique.

8. What are the inclusion and exclusion criteria of the study? Not clearly mentioned or I think it is missed from the manuscript?

 Of course, we have seen that we missed the inclusion and exclusion criteria of the study in our submitted manuscript so that we incorporated in the revised manuscript.

 Inclusion and exclusion criteria of the study

 Inclusion Criteria

All nurses who were fulltime employees of Harari regional state governmental hospitals were included. 

 Exclusion Criteria

Nurses who were seriously ill or unable to give responses due to this illness during the data collection period were excluded.

9. What is your assumption with the outcome variables from the study units or across hospitals is it similar or different?

 We assumed that the level of work-related stress across each hospital is different due to the difference in the population they served so that a stratified random sampling technique was used to select study participants from each hospital to maintain proportionality among each hospital.

10. Variable measurement or operational definition Missed.

 Operational Definitions:

Work-related stress: is defined as the physical and emotional reactions that occur when the nurses' abilities and resources imbalance with the demands and requests of their work.

Nurses' work-related stress: was rated from 1 (never stressful) to 4 (always stressful); the mean score of 2 or higher of 57 items was considered as work-related stress. 

Substance use: use or consumption of any substance such as alcohol, chat, cigarettes, shisha, and hashish, regardless of the amount and frequency of use for the past 3 months.

Job satisfaction: Job Satisfaction was defined as whether respondents like (satisfied) or dislike (dissatisfied) their jobs. In this study, nurses who a¬swered “yes” were assumed to be satisfied with their job.

Chronic medical illness: ‘Chronic medical disease’ was categorized as ‘yes’ if hypertension, cardiovascular disease, AIDS, diabetes mellitus or arthritis had been diagnosed.

11. What is Cronbach’s alpha coefficient implications and its cut off point for conclusion?

 Cronbach’s alpha coefficient is a measure of internal consistency.

 It is considered to be a measure of scale reliability.

 A reliability coefficient of 0.70 or higher is considered acceptable for conclusion. 

12. Was the tools used for the study were, the primary justifications for variations of different findings with yours, what are other possible expectations or perspectives behind for the difference can you suggest?

 In fact, the cause of stress in an individual person is difficult to determine since the stressors may vary from culture to culture even nurses working in different hospital units exposed to different types of work-related stress. Therefore, the variations for the level of stress among study findings might be due to the difference in the working environment itself or other factors in addition to the tools used to measure the stress level. So, further qualitative research could be used to explore and describe the experiences of stress among nurses in the work environment. 

13. Making the general conclusion with the descriptive result is not recommended.so try to remove this sentence "The descriptive analysis showed that the workload was the most frequently stressful sub-scale.”

 We accepted it and removed it in the revised manuscript.

14. The manuscript has no page numbers. The editor relies on whole article pagination as downloaded from the website. Assuming that the Abstract is on page 8.

 Sorry for the omission unintentionally and we have inserted page numbers and line numbers for each page of the manuscript.

15. Please include socio-demographics in abstract results.

 We accepted and did it in the revised manuscript.

16. The sampling section was not clearly written. Please revisit. The author starts with simple random sampling but ends up with stratified random sampling but provides numbers for each stratum. The sampling frame is small (446). Why was a census not considered?

 We have indicated how to select study samples in question number 7 of this letter.

 It could be sounder enough if we had applied census for the study but we did not have adequate resources and time to apply census during the study period and we also considered that the sample size is acceptable for this study.

17. Keep reporting of numbers to 1 decimal place.

 We accepted and did it in the revised manuscript.

---

## [Decision Letter · Decision Letter 1]

4 Feb 2020

PONE-D-19-14479R1

Nurses’ work-related stress and associated factors in governmental hospitals in Harar, Eastern Ethiopia: A cross-sectional study

PLOS ONE

Dear MR. Baye,

Thank you for submitting your manuscript to PLOS ONE. After careful consideration, we feel that it has merit but does not fully meet PLOS ONE’s publication criteria as it currently stands. Therefore, we invite you to submit a revised version of the manuscript that addresses the points raised during the review process.

We would appreciate receiving your revised manuscript by Mar 20 2020 11:59PM. To enhance the reproducibility of your results, we recommend that if applicable you deposit your laboratory protocols in protocols.io, where a protocol can be assigned its own identifier (DOI) such that it can be cited independently in the future. For instructions see: http://journals.plos.org/plosone/s/submission-guidelines#loc-laboratory-protocols

We look forward to receiving your revised manuscript.

Kind regards,

Sphiwe Madiba, DrPH

Academic Editor

PLOS ONE

Additional Editor Comments 

Please correct the sampling technique uses, in the abstract you stated that simple random sampling was used and in the method section, you mention stratified random sampling. The study population was 442 and the sample 398, I do not understand how simple random sampling was used when almost all the nurses were included in the study. How did you randomly select 217 from 244 nurses, 96 from 97 nurses and 50 from 55 nurses? The appropriate sampling technique for the study would be a census were all the nurses were eligible to participate. Provide details or change the sampling to a census and clarify how the participants were recruited to the study.Delete the sub heading “Inclusion and exclusion criteria” and integrate the description of the population under study populationFigure, “distribution of study participants by the hospital, is not necessary, delete and present the data in the narrativePlease explain child rearing

**Discussion**

Make this the second statement after the prevalence. *The descriptive analysis indicated that the workload was the most 298 stressful subscale for nurses.* Tell us why this is the case and the implications of the findings before telling us that- this was consistent with other studies in different countries [(2717, 299 36-4624-34]). This might be due to a shortage of staffs, extra non-nursing tasks and less time to support each other emotionally-support this statement with literature. Only then can you compare the prevalence with other studies. Please be brief in you comparison and group all the studies that are similar then those that are in contrast.

The implication should be part of the conclusion and not a subheading-see my comments above.

**Conclusion**

Child rearing is not discussed but appears in the conclusion

There are still major grammar, tenses, and punctuation errors-the manuscript should be copy edited by a professional English language editor. Please provide proof with your re-submission.

Reviewers' comments:

Reviewer's Responses to Questions

**Comments to the Author**

1. If the authors have adequately addressed your comments raised in a previous round of review and you feel that this manuscript is now acceptable for publication, you may indicate that here to bypass the “Comments to the Author” section, enter your conflict of interest statement in the “Confidential to Editor” section, and submit your "Accept" recommendation.

Reviewer #1: All comments have been addressed

Reviewer #2: All comments have been addressed

2. Is the manuscript technically sound, and do the data support the conclusions?

Reviewer #1: Yes

Reviewer #2: Yes

3. Has the statistical analysis been performed appropriately and rigorously? 

Reviewer #1: Yes

Reviewer #2: Yes

4. Have the authors made all data underlying the findings in their manuscript fully available?

Reviewer #1: Yes

Reviewer #2: Yes

5. Is the manuscript presented in an intelligible fashion and written in standard English?

Reviewer #1: Yes

Reviewer #2: Yes

6. Review Comments to the Author

Reviewer #1: Dear academic Editor

I try to check for author’s response to my review comments and it is well done.

I agreed all the responses given that was constructive and educative responses and help to improve the quality of the paper. 

Reviewer #2: The comments and suggestions that I raised in the review have been adequately attended to by the authors.

7. PLOS authors have the option to publish the peer review history of their article (what does this mean?). If published, this will include your full peer review and any attached files.

Reviewer #1: Yes: Getnet Gedif

Reviewer #2: No

---

## [Author Response · Author response to Decision Letter 1]

20 Mar 2020

Response to Reviewers’ 

First of all, we would like to appreciate the PLOSE ONE Journal Academic Editor and Reviewers for providing us timely and constructive comments to revise our manuscript that suits the PLOS ONE Journal standards and requirements. We are also pleased to submit the revision that responds to each point raised by the academic editor/ reviewers and presented as follows.

1. Please correct the sampling technique uses, in the abstract, you stated that simple random sampling was used and in the method section, you mention stratified random sampling. 

 As you mentioned, we have seen that stratified random sampling was reported in the methods section and simple random sampling both in the abstract and in the methods section. A stratified sampling technique was used to proportionally allocate the number of participants to be involved in the study by considering the difference in the level of work-related stress across hospitals. Then we used a simple random sampling technique to select study participants from each hospital.

2. The study population was 442 and the sample 398, I do not understand how simple random sampling was used when almost all the nurses were included in the study. How did you randomly select 217 from 244 nurses, 96 from 97 nurses and 50 from 55 nurses? The appropriate sampling technique for the study would be a census were all the nurses were eligible to participate. Provide details or change the sampling to a census and clarify how the participants were recruited to the study.

 The study population was 446 and the sample 398. Out of 446 study population, HUSTH comprised 244 nurses and JH had 97 nurses. Similarly, there were 55 nurses in AH and 50 nurses in PH. As we have indicated in the methods section, the total sample size was proportionally allocated to the size of the four government hospitals so that HUSTH, JH, AH and PH constituted 217, 87, 49 and 45 participants respectively. Then, the sampling frame was constructed for each hospital separately by taking the nurses list from the human resource management department of each hospital and study participants were selected from respective hospitals using a simple random sampling technique.

 In fact, It could be sounder enough if we had applied census for the study but we did not have adequate resources and time to apply census during the study period and we also considered that the sample size was acceptable for this study.

3. Delete the subheading “Inclusion and exclusion criteria” and integrate the description of the population under study population

 We accepted and corrected it in the revised manuscript.

4. Figure,“distribution of study participants by the hospital, is not necessary, delete and present the data in the narrative

 We accepted and corrected it in the revised manuscript.

5. Please explain child rearing

 Child-rearing: Child rearing was categorized as ‘yes’ if the person is parenting or taking care of a child/children

 We also incorporated it under operational definition of variables in the revised manuscript.

6. Make this the second statement after the prevalence. The descriptive analysis indicated that the workload was the most 298 stressful subscale for nurses. Tell us why this is the case and the implications of the findings before telling us that- this was consistent with other studies in different countries [(2717, 299 36-4624-34]). This might be due to a shortage of staff, extra non-nursing tasks and less time to support each other emotionally-support this statement with literature. Only then can you compare the prevalence with other studies. Please be brief in your comparison and group all the studies that are similar then those that are in contrast.

 We accepted and corrected it in the revised manuscript.

7. The implication should be part of the conclusion and not a subheading

 We accepted and corrected it in the revised manuscript.

8. Child-rearing is not discussed but appears in the conclusion

 We have seen and corrected it in the revised manuscript.

9. There are still major grammar, tenses, and punctuation errors-the manuscript should be copy edited by a professional English language editor. Please provide proof with your re-submission.

 We tried to improve the manuscript for grammar, tenses, and punctuation errors by consulting our colleagues and staff language professionals.

---

## [Decision Letter · Decision Letter 2]

27 May 2020

PONE-D-19-14479R2

Nurses’ work-related stress and associated factors in governmental hospitals in Harar, Eastern Ethiopia: A cross-sectional study

PLOS ONE

Dear Dr. Baye,

Thank you for submitting your manuscript to PLOS ONE. After careful consideration, we feel that it has merit but does not fully meet PLOS ONE’s publication criteria as it currently stands. Therefore, we invite you to submit a revised version of the manuscript that addresses the points raised during the review process.

Your paper has been reviewed again by the academic referees performing the first assessment, having a positive appraisal. nevertheless, one of these reviewers requests for a set of minor changes before suggesting the acceptance of the manuscript in PLOS ONE. Specifically, there are crucial issues related to 1) the sampling procedure and its validity, that remains pending to be clarified, and 2) the English writing of the paper, that needs additional revisions, to be (ideally) carried out through a professional reading-proof.

We look forward to receiving your revised manuscript.

Kind regards,

Sergio A. Useche, Ph.D.

Academic Editor

PLOS ONE

Reviewers' comments:

Reviewer's Responses to Questions

**Comments to the Author**

1. If the authors have adequately addressed your comments raised in a previous round of review and you feel that this manuscript is now acceptable for publication, you may indicate that here to bypass the “Comments to the Author” section, enter your conflict of interest statement in the “Confidential to Editor” section, and submit your "Accept" recommendation.

Reviewer #1: All comments have been addressed

Reviewer #2: (No Response)

2. Is the manuscript technically sound, and do the data support the conclusions?

Reviewer #1: Yes

Reviewer #2: Yes

3. Has the statistical analysis been performed appropriately and rigorously? 

Reviewer #1: Yes

Reviewer #2: Yes

4. Have the authors made all data underlying the findings in their manuscript fully available?

Reviewer #1: Yes

Reviewer #2: No

5. Is the manuscript presented in an intelligible fashion and written in standard English?

Reviewer #1: Yes

Reviewer #2: No

6. Review Comments to the Author

Reviewer #1: Dear Academic Editor

Confirmation to the authors’ response to the reviewer 1 Comments

Title: Nurses’ work-related stress and associated factors in governmental hospitals in Harar, Eastern Ethiopia:

A cross-sectional study"

Manuscript number: (PONE-D-19-14479R2 )

I tried to check out authors' response to my review comments and it is well done.

I agreed to all the responses' given which was constructive and educative in addition to improve

quality of the paper.

I have re-checked and confirmed all the authors’ response

to my review comments that was fully a dressed.

Reviewer #2: The issues of sampling is not adequately addressed. One of the reviewer comments was adequate. That the sampling likely followed a census rather than random sampling. It is difficult to understand how one can obtain a random sample of 398 from a population of 446 people using the technique given in manuscript.

Also there are still grammatical and typographical errors in the manuscript. These still to be addressed.

7. PLOS authors have the option to publish the peer review history of their article (what does this mean?). If published, this will include your full peer review and any attached files.

Reviewer #1: Yes: Getnet Gedif

Reviewer #2: No

---

## [Author Response · Author response to Decision Letter 2]

9 Jun 2020

Response to Reviewers’ 

First of all, we would like to appreciate the PLOSE ONE Journal Academic Editor and Reviewers for providing us timely and constructive comments to revise our manuscript that suits the PLOS ONE Journal standards and requirements. We are also pleased to submit the revision that responds to each point raised by the academic editor/ reviewers and presented as follows.

1. The sampling procedure and its validity, that remains pending to be clarified.

 The calculated sample size (398) was proportionally allocated to four government hospitals (HUSTH, JH, AH, and PH) as indicated in the methods section of the revised manuscript.

 Then, the sampling frame was constructed for each hospital separately by taking the nurses list from the human resource management department of each hospital and study participants were selected from respective hospitals using simple random sampling technique.

 We have reported what we have done in the study.we considered that the sampling procedure followed acceptable scientific standards and the sample size was adequate for this study.

2. The English writing of the paper, that needs additional revisions. 

 We made extensive revisions to the revised manuscript to improve grammar, tenses, and punctuation errors by consulting our colleagues and staff English language professionals.

---

## [Decision Letter · Decision Letter 3]

15 Jul 2020

Nurses’ work-related stress and associated factors in governmental hospitals in Harar, Eastern Ethiopia: A cross-sectional study

PONE-D-19-14479R3

Dear Dr. Baye,

We’re pleased to inform you that your manuscript has been judged scientifically suitable for publication and will be formally accepted for publication once it meets all outstanding technical requirements.

Kind regards,

Sergio A. Useche, Ph.D.

Academic Editor

PLOS ONE

Additional Editor Comments (optional):

Reviewers' comments:

Reviewer's Responses to Questions

**Comments to the Author**

1. If the authors have adequately addressed your comments raised in a previous round of review and you feel that this manuscript is now acceptable for publication, you may indicate that here to bypass the “Comments to the Author” section, enter your conflict of interest statement in the “Confidential to Editor” section, and submit your "Accept" recommendation.

Reviewer #2: All comments have been addressed

2. Is the manuscript technically sound, and do the data support the conclusions?

Reviewer #2: Yes

3. Has the statistical analysis been performed appropriately and rigorously? 

Reviewer #2: Yes

4. Have the authors made all data underlying the findings in their manuscript fully available?

Reviewer #2: Yes

5. Is the manuscript presented in an intelligible fashion and written in standard English?

Reviewer #2: Yes

6. Review Comments to the Author

Reviewer #2: I would have wanted the authors to motivate why they could not do a census rather than sampling given that the sample size and the sampling frame numbers are close.

7. PLOS authors have the option to publish the peer review history of their article (what does this mean?). If published, this will include your full peer review and any attached files.

Reviewer #2: **Yes: **Prof Paul Chelule

---

## [Editor Report · Acceptance letter]

23 Jul 2020

PONE-D-19-14479R3 

Nurses’ work-related stress and associated factors in governmental hospitals in Harar, Eastern Ethiopia: A cross-sectional study 

Dear Dr. Baye:

I'm pleased to inform you that your manuscript has been deemed suitable for publication in PLOS ONE. Congratulations! Your manuscript is now with our production department. 

Kind regards, 

on behalf of

Dr. Sergio A. Useche 

Academic Editor

PLOS ONE